# Organophosphorus Flame Retardant TDCPP Displays Genotoxic and Carcinogenic Risks in Human Liver Cells

**DOI:** 10.3390/cells11020195

**Published:** 2022-01-07

**Authors:** Quaiser Saquib, Abdullah M. Al-Salem, Maqsood A. Siddiqui, Sabiha M. Ansari, Xiaowei Zhang, Abdulaziz A. Al-Khedhairy

**Affiliations:** 1Zoology Department, College of Sciences, King Saud University, P.O. Box 2455, Riyadh 11451, Saudi Arabia; alsalem1985@hotmail.com (A.M.A.-S.); maqsoodahmads@gmail.com (M.A.S.); kedhairy@ksu.edu.sa (A.A.A.-K.); 2Botany and Microbiology Department, College of Sciences, King Saud University, P.O. Box 2455, Riyadh 11451, Saudi Arabia; sabiha.mahmood003@gmail.com; 3State Key Laboratory of Pollution Control & Resource Reuse, School of the Environment, Nanjing University, Nanjing 210023, China; zhangxw@nju.edu.cn

**Keywords:** TDCPP, flame retardants, genotoxicity, hepatotoxicity, toxicogenomics, apoptosis

## Abstract

Tris(1,3-Dichloro-2-propyl)phosphate (TDCPP) is an organophosphorus flame retardant (OPFR) widely used in a variety of consumer products (plastics, furniture, paints, foams, and electronics). Scientific evidence has affirmed the toxicological effects of TDCPP in in vitro and in vivo test models; however, its genotoxicity and carcinogenic effects in human cells are still obscure. Herein, we present genotoxic and carcinogenic properties of TDCPP in human liver cells (HepG2). 3-(4,5-Dimethylthiazol-2-yl)-2,5-diphenyl-2H-tetrazolium bromide (MTT) and neutral red uptake (NRU) assays demonstrated survival reduction in HepG2 cells after 3 days of exposure at higher concentrations (100–400 μM) of TDCPP. Comet assay and flow cytometric cell cycle experiments showed DNA damage and apoptosis in HepG2 cells after 3 days of TDCPP exposure. TDCPP treatment incremented the intracellular reactive oxygen species (ROS), nitric oxide (NO), Ca^2+^ influx, and esterase level in exposed cells. HepG2 mitochondrial membrane potential (*ΔΨm*) significantly declined and cytoplasmic localization of P53, caspase 3, and caspase 9 increased after TDCPP exposure. qPCR array quantification of the human cancer pathway revealed the upregulation of 11 genes and downregulation of two genes in TDCPP-exposed HepG2 cells. Overall, this is the first study to explicitly validate the fact that TDCPP bears the genotoxic, hepatotoxic, and carcinogenic potential, which may jeopardize human health.

## 1. Introduction

The production of an alternative to flame retardants, known as organophosphorus flame retardants (OPFRs), has swiftly increased, allowing it to be used in a range of commercial products [1]. OPFRs are widely used in household and commercial items, including plastics, furniture, paints, foams, electronic goods, building materials, and construction materials [2]. A large amount of scientific evidence has affirmed that OPFRs can have adverse effects on humans and animals [3,4,5,6]. Tris(1,3-dichloro-2-propyl)phosphate (TDCPP) is an OPFR that is frequently added in polyvinyl chloride, rigid polyurethane foam, epoxy resin, and polyester fiber synthesis. Nonetheless, TDCPP is one of the most common OPFRs used in baby products [7]. Infants are at high risk of exposure to TDCPP owing to the fact that TDCPP and its metabolites have been detected at a significantly higher level in their urine samples [8]. Recent and earlier reports have confirmed the presence of TDCPP in house dust samples [9,10]. Around 96% of dust samples collected from USA households showed >2 to 50 ppm of TDCPP [11]. Dermal absorption and hand-to-mouth contact have been attributed to be a key factor for the entry of TDCPP into the human body [7]. Based on the two highest indoor dust levels of TDCPP, oral doses determined for pregnant women were 64.2 μg/day and 9.0 μg/day, which exceed the non-significant risk levels for cancer of 5.4 μg/day [12]. Human biomonitoring studies have confirmed the presence of TDCPP in human breast milk (162 ng/g lipid weight), adipose tissues (260 ng/g), and urine (metabolite BDCPP, 62.1−1760 pg/mL), as well as a 3% decrease in free thyroxin and 17% increase in prolactin [13,14,15,16,17]. Fire fighters encountering items containing OPFRs have demonstrated higher level of BDCPP, a metabolite of TDCPP, in their urine, indicating their high risk to absorb OPFRs [18].

TDCPP not only endangers human health; in fact, its concentration buildup in the environment raises alarming concerns. Samples taken from a Japanese sea-based solid waste disposal site also showed TDCPP in them [19]. Seawater collected from China, Germany, and USA (California) demonstrated the presence of 109.28, 50, and 1.4 × 10^3^ ng/L of TDCPP [20,21,22]. Laundry wastewater from USA households was also detected with (up to 65,600 ng/L) high levels of TDCPP [23]. Benthic invertebrates (bivalve, snail, and mussel) showed 15% TDCIPP, while 12.5% TDCPP was detected in shrimps [24]. Moreover, wild biota showed the presence of TDCPP ranging from <1.1 to 140 ng/g lw (lipid weight) [25]. Norwegian great black-backed gull eggs and Great Lakes herring gull eggs were detected with the presence of TDCPP ranging between 1.9 and 0.17 ng/g ww [26]. Freshwater fish showed 251 μg/kg lw TDCPP in muscles [27]. Apart from the environmental occurrence and bioaccumulation, under the laboratory conditions, TDCPP-exposed zebrafishes showed its accumulation in brain tissues, which induced endocrine disruption and reproductive changes, swim bladder defects, enzymatic imbalances, cell cycle arrest, DNA damage, and transcriptional activation of oxidative stress genes [28,29,30,31,32]. Zebrafishes also showed hormonal alterations, and embryonic toxicity recorded as defects in gastrulation and aberrant germ layer formation, which leads to the development of abnormal tissue and organs after TDCPP exposure [33,34]. Chicks exposed to TDCPP for 21 days showed degenerate Purkinje cells, signifying the interruption of their neurodevelopment processes [35]. Besides, TDCPP has disrupted the gut microbiome and related gene expression in mice [36].

In vitro studies on human cells exposed to TDCPP have also demonstrated its acute toxicity in hepatocytes and neuronal cells [3]. PC12 cells exposed to TDCPP inhibited the cell proliferation and caused morphological changes, altered the expressions of genes and proteins, and triggered apoptosis [37]. HUVECs treated with TDCPP-induced vascular toxicity through the Nrf2-VEGF pathway [38]. HK-2 cells treated with low concentrations of TDCPP demonstrated cytostasis [39]. HaCaT cells treated with TDCPP showed toxicological effects manifested as cell cycle arrest and apoptotic cell death [40]. TDCPP has been reported to induce neurotoxicity in pheochromocytoma neuronal cells and neuroblastoma cells by elevating the intracellular Ca^2+^ level, and simultaneous disruption of signaling pathways [41]. Human lung cells (A549) grown in the presence of TDCPP exhibited a decline in cell survival with concomitant effects on DNA damage, mitochondrial and cell cycle dysfunctions [42]. Rat pituitary cell line (GH3) exposed to TDCPP and its metabolite BDCPP demonstrated cytotoxicity, as well as thyroid hormone-mimicking effects via a strong binding with the membrane receptor and activation of the ERK1/2 pathway [43].

In vitro mouse liver microsome metabolomic analysis revealed the fact that TDCPP can easily bio-accumulate, as compared to other OPFRs [44]. Hepatocytes (AML-12 cells) exposed to TDCPP preferentially targeted the mitochondria and disturbed its morphology and function (potential, metabolism, and elevated mito-ROS), as well as caused lipid accumulation at non-cytotoxic concentration [45]. Human hepatocarcinoma cells (MMC-7721) exposed to TDCPP stimulated an imbalance in the oxidative enzymes (SOD, LDH, and CAT), which leads to apoptosis in cells [46]. Rats, avian liver cells, and American kestrel exposed to TDCPP exhibited increased liver weight, disturbances in fat metabolism, and liver integrity [3,47,48]. In the same line, HepG2 cells exposed to TDCPP have been linked with uproars in lipid metabolism and cholesterol biosynthesis [47,48]. The above studies unequivocally pointed towards the liver toxicity of TDCPP. Despite the prominent role of liver as a foremost organ to encounter and metabolize environmental toxicants, we did not find any study on TDCPP that unraveled its genotoxic and carcinogenic potential in human liver cells. Hence, we specifically used HepG2 cells owing to its greater ability for differentiation and genotypic similarities with the normal human liver cells. Consequently, due to such attributes of HepG2, it has been a preferred cell line for the toxicological screening of chemicals [49]. Hence, this study is a combination of traditional toxicological (cytotoxicity, DNA damage, oxidative stress, and apoptosis) and molecular approaches (activation of apoptotic proteins, qPCR array of cancer pathway genes) to elucidate the cellular mechanism of hepatotoxicity and carcinogenicity of TDCPP in HepG2 cells.

## 2. Materials and Methods

### 2.1. Cell Survival Analysis

The effects of tris(1,3-dichloro-2-propyl)phosphate (TDCPP) (CAS # 13674-87-8, Cat # 32951, Sigma-Aldrich, St. Louis, MO, USA) on HepG2 survival were quantitated by MTT and NRU assays following a previously described method [50]. Exposure concentrations of TDCPP were selected based on screening tests performed for 24–72 h with a range of concentrations (5, 10, 25, 50, 100, 200, and 400 µM). However, up to 50 µM, no significant cytotoxic effects were observed after 72 h. Only higher concentrations (i.e., 100, 200, and 400 µM) demonstrated significant cytotoxicity after 72 h (Appendix A). The tested concentrations in our study is nearby to the realistic TDCPP concentrations detected in the indoor dust (<0.03 to 326 mg/g) [14]. Moreover, previous studies on TDCPP have also verified its toxicity in mammalian cells ranging from ≥100 μg/mL for 48 h [40] and 250 μM [39]. Cell survival was quantitated by the MTT assay, which relies on the reduction of water-soluble tetrazolium into an insoluble formazan product by the live cell mitochondrial dehydrogenase enzyme [51]. Briefly, HepG2 cells were grown in 96-well plates at 37 °C overnight in a CO_2_ incubator (5% and 95% humidity). Afterwards, cells were exposed to 100–400 μM of TDCPP for 72 h in RPMI-1640 medium (without FBS). Post exposure, the medium was aspirated and washed with PBS, and 10 µL MTT (5 mg/mL) was added, and left for 4 h incubation. Finally, the medium was decanted, 200 µL/well DMSO was added, and the solution was pipetted gently, followed by absorbance measurement at 550 nm on a microplate photometer (Multiskan Ex, Thermo Fisher Scientific, Vantaa, Finland). The NRU assay is based on the lysosomal accumulation of neutral red dye by the live cells; disturbances or fragility in its membrane ultimately affects the above process [52]. In the case of the NRU assay, HepG2 cells were exposed in the same way as described above. Afterwards, the medium was discarded, dye (NR; 50 μg/mL) was added to each well, and left for 3 h incubation. Subsequently, unbound dye was washed using a 1% CaCl_2_ and 0.5% CH_2_O solution, followed by the addition of 200 µL/well of dilution buffer (1% acetic acid and 50% ethanol). The solutions in the wells were gently pipetted and the absorbance was recorded at 550 nm on a microplate photometer.

### 2.2. Analysis of DNA Damage by Comet Assay

Comet assay provides an upper hand in measuring the DNA damage in an individual cell. Under alkaline conditions, the broken DNA is electrostretched from the nuclei head, forming a comet tail that can be measured as DNA damage in cells. DNA damage in HepG2 cells was examined by a previously described method [53,54]. TDCPP-exposed cells (100–400 µM, 3 days) were trypsinized and washed twice with PBS at 3000 rpm for 3 min. Cell pellets were suspended in 100 µL PBS and 100 µL low melting agarose. Comet slides were prepared by aspirating a volume of 80 µL and spread on a glass slide (1/3rd frosted) using a cover slip (No.1). The slides were then kept on ice for gel solidification. Normal melting agarose (90 µL) was overlaid on the solidified gel, and again left on ice for gel solidification. All slides were kept at 4 °C overnight in a lysing solution. Subsequently, all slides were subjected to DNA unwinding (20 min) and electrophoresis at 24 V (30 min). Staining of DNA was done with 20 µg/mL of EtBr. DNA damage was analyzed under a fluorescence microscope using comet assay IV software (Instem, Perceptive Instruments, Cambridge, UK).

### 2.3. Cell Death and Oxidative Stress Quantification

Flow cytometry gave the freedom to analyze the activity of cells (fluorochromes stained) passing in a row from the laser beam, whose scattering patterns are captured by the optics and processed by electronics to generate data. We have analyzed the cell death and oxidative stress in TDCPP-exposed cells by flow cytometry as stated earlier [55]. After the exposure to TDCPP (100–400 μM, 3 days), cells were detached and pelleted at 3000 rpm for 3 min. For cell death, cells were fixed in 70% ethanol (500 μL) for 1 h. Afterwards, the cells were washed with PBS and stained with propidium iodide (500 μL). Ten thousand cells were analyzed for cell death by measuring the cell cycle on a flow cytometer (Coulter Epics XL/XL-MCL, Miami, FL, USA). For the oxidative stress analysis, cells were exposed to the same condition as described above. Afterwards, cells were detached and washed with PBS. For ROS analysis, cell pellets were stained with DCFH-DA (5 μM); for NO analysis, staining of cells was performed with DAF2-DA (5 µM) for 1 h. Intracellular fluorescence intensities of dyes were quantitated in ten thousand cells on a flow cytometer.

### 2.4. MMP, Ca^++^ Influx, and Cellular Esterase Quantification

HepG2 cells were grown in the presence of TDCPP (100–400 μM) for 3 days. The cells were harvested by trypsinization and washed with PBS. Three cell suspensions were prepared in PBS (500 μL) and 1 h staining was performed with 5 µg/mL of Rh123 (MMP analysis), 4 µM of Fluo-3 (Ca^++^ influx), and 5 µM of CFDA-SE dyes. Post staining, intracellular fluorescence intensities were quantitated in ten thousand cells on a flow cytometer [56].

### 2.5. Immunolocalization Analysis

The immunofluorescence assay utilizes the specificity of antibodies tagged with a fluorescence dye to recognize their antigen and helps in visualizing the distribution of the target molecule under a fluorescence microscope. HepG2 cells were seeded on a glass chamber slide and treated with TDCPP (100 μM, 3 days). Post exposure, the cells were thoroughly washed with PBS and fixed in 70% methanol. Blocking was performed with 5% BSA. Afterwards, the cells were incubated for 60 min in a 1:50 dilution of 1°Ab of P53, caspase 3, and caspase 9 (Santa Cruz Biotechnology Inc., Dallas, TX, USA). Subsequently, goat anti-rabbit IgG-TR and goat anti-rabbit IgG-FITC antibodies (2°Ab) were used to incubate the cells in a dilution of 1:100 for 1 h. Nuclear staining was performed with 600 nM DAPI, and images were taken using a fluorescence microscope (Nikon Eclipse 80i, Tokyo, Japan) [56].

### 2.6. Transcriptomic Analysis

RT^2^ Profiler PCR Array is a reliable tool for the quantification of transcriptional variations in the panel of genes belonging to a specific pathway using a real-time PCR machine. TDCPP (100 μM, 3 days)-treated and control cells were harvested and washed with PBS. Total RNA was isolated using RNeasy Mini kit and purified using a PureLink kit, and quantification was performed on Nanodrop 8000 (Thermo Fisher, Waltham, MA, USA). cDNA was synthesized from 1 μg of total RNA using ready-to-go PCR beads (GE Healthcare, UK). Transcriptomic variations in 84 genes (human cancer pathway) were analyzed in 96-well array plates (PAHS-033Z, SA Biosciences Corporation, Frederick, MD, USA) on a real-time PCR machine (Roche LightCycler^®^ 480, Basel, Switzerland) [57]. Data are presented as heat map and scatter plot determined after normalizing the expression of genes with endogenous genes (*ACTB*, *RPLP0*, *HPRT1*, *B2M*, and *GAPDH*) using the manufacturer’s Gene Globe online software (Qiagen, Germantown, MD, USA).

## 3. Results

### 3.1. TDCPP Induces Cytotoxicity

Compared to control cells, HepG2 grown in the presence of TDCPP exhibited loss in population density and cellular shrinkage. Under similar conditions, the control cells showed characteristic epithelial cell morphology (Figure 1A). Quantitative analysis of TDCPP (100–400 μM) cytotoxicity measured using the MTT assay exhibited 16.2 ± 0.92%, 31.5 ± 1.32% and 68.0 ± 1.09% declines in the survival of HepG2 cells (Figure 1B). The IC50 of TDCPP determined from the MTT assay is 163.2 μM (LogIC50 2.213, R^2^ 0.9165). The lysosomal toxicity analysis by the NRU assay resulted in a significant reduction of cell survival to 10.81 ± 0.91%, 25.58 ± 2.17%, and 63.62 ± 1.53% after TDCPP (100–400 μM) exposure (Figure 1C).

### 3.2. Genotoxicity of TDCPP

Comet assay results of TDCPP-exposed HepG2 cells showed a considerable number of DNA strand breaks (appearing as tail), compared to their respective controls (Figure 2). Quantitative analysis showed significant DNA damage in exposed cells. Compared to 0.29 OTM in the control, HepG2 cells grown for 3 days in the presence of 100, 200, and 400 μM of TDCPP showed 6.55-, 16.58-, and 30.58-fold greater OTM values (Appendix A). Frequency distribution graphs exhibit different levels of DNA damage in the population of HepG2 cells after treatment with TDCPP (Appendix A).

### 3.3. TDCPP Triggers Apoptosis and Induce Oxidative Stress

Flow cytometric results display an increase in the apoptotic peak (subG1) in HepG2 cells after 3 days of TDCPP (100–400 μM) exposure (Figure 3A). Heat maps show the mean data of HepG2 cell cycle phases (G1, S, G2M, SubG1) recorded after TDCPP (100–400 μM) exposure. Referring to control subG1 (6.1%), TDCPP exposure showed significantly higher (49.7%, 64.4%, and 70.8%) subG1 cell populations in apoptotic phase (Figure 3B).

Flow cytometric results showed an increment in the intracellular ROS and NO levels after TDCPP exposure. Illustrative overlay graphs of TDCPP (100, 200, and 400 μM) treated cells showed 19.8%, 17.5%, 15.2% and 13.0%, 13.9%, 15.1% MnIX of DCF and DAF-2DA vis-à-vis their controls showed MnIX of 9.0% and 12.1% (Figure 4A,C). The mean data of ROS and NO in TDCPP (100–400 μM)-exposed cells demonstrated significantly higher (220.11%, 194.44%, 168.88% and 13.0%, 13.9%, 15.1%) ROS and NO production in cells (Figure 4B,D).

### 3.4. TDCPP Disrupts MMP and Elevates Ca^++^ and Cellular Esterase Levels

TDCPP (100–400 μM) exposure reduced the MMP (*ΔΨ*) in HepG2 cells. The flow cytometric overlay graph measured as Rh123 fluorescence exhibited 10.5%, 9.3%, and 8.9% MnIX, compared to 12.5% in the control (Figure 5A). Relative to the control, the mean data of TDCPP (100–400 μM) treated cells exhibited significant reductions of Rh123 fluorescence to 16.5%, 25.0%, and 28.9% (Figure 5B).

HepG2 cells treated with TDCPP (100–400 μM) caused Ca^++^ influx, measured as Fluo-3 fluorescence enhancement. The representative overlay graph showed MnIX of 16.6%, 18.3, and 24.4% in TDCPP (100–400 μM) treated cells, while the control showed 9.9% MnIX (Figure 5C). The mean data of MnIX exhibited significantly greater (170.6%, 187.4%, and 246.7%) Ca^++^ influx in cells over its control (Figure 5D).

TDCPP (100–400 μM) exposure resulted in the elevation of the cellular esterase level, recorded as CFDA-SE fluorescence increase. Compared to the MnIX of 5.6% in the control, TDCPP-exposed cells showed a gradational shift of MnIX values (7.3%, 11.7%, 15.9%) (Figure 5E). Relative to control, HepG2 cells treated with 100–400 μM of TDCPP exhibited 131.4%, 205.8%, and 279.7% greater esterase levels (Figure 5F).

### 3.5. TDCPP Activated DNA Damage and Apoptotic Proteins

TDCPP exposure activated the immunolocalization of P53 and caspase 3 and 9 proteins in the cells. Compared to the control, TDCPP (100 μM) exposure resulted in the cytoplasmic and nucleolar localization of P53 (Figure 6). TDCPP (100 μM) exposed cells also demonstrated high fluorescence caspases 3 and 9 in the cytoplasm (Figure 6).

### 3.6. TDCPP Triggers Cancer Pathway Genes

RT^2^ profiler PCR array of TDCPP (100 μM) treated cells exhibited upregulation and downregulation of cancer pathway genes in HepG2 cells, depicted as a heat map and its corresponding list of genes in the qPCR array plate (Figure 7A). The scatter plot depicts the normalized expression of all genes between TDCPP and the control (Figure 7C). Within the group of over-expressed genes (>2.0-fold) are *CCND3* (2.13-fold), *GADD45G* (119.43-fold), *GSC* (2.04-fold), *IGFBP5* (8.34-fold), *PFKL* (3.56-fold), *SKP2* (3.73-fold), *SNAI2* (2.57-fold), *TEK* (2.83-fold), *TERF2IP* (2.17-fold), *TNKS2* (7.67-fold), and *UQCRFS1* (3.61-fold). Two genes that were under-expressed are *SNAI1* (−3.51-fold) and *TEP1* (−2.79-fold) (Figure 7B).

## 4. Discussion

TDCPP has been commercially used in several consumer products; numerous studies have documented its existence in indoor air and dust, workplaces, and various environmental locations [58]. In fact, Quick Chemical Assessment Tool has listed TDCPP as a Cat I* compound, indicating its high toxicological concern [59]. In recent years, humans living in different geographical locations have also shown the body burden of TDCPP [15,60,61,62]. In spite of this fact, we have not found studies on the genotoxic and hepatotoxic potential of TDCPP in human liver cells. Consequently, understanding hepatotoxic effects of TDCPP will help in defining its risks to the average consumers and first responders to suppress the incidences of fire.

Cell viability analysis is crucial to evaluate the sensitiveness of a cell to toxicants, as it mirrors its well-being and ability to survive [3]. Hence, in this connection, our study is a first report on the cyto- and genotoxicity of TDCPP in human liver cells (HepG2). We have found that TDCPP did not reduce the cell viability in MTT and NRU until after exposure to 100–400 μM for 3 days. Cell viability significantly declined in both assays explicitly affirm the effect of TDCPP on the malfunctioning of mitochondrial dehydrogenase enzyme and lysosomal membrane integrity [57]. Our cytotoxicity data corroborate well with previously published reports on human cell lines (H295R, PC12, HaCaT, and HepG2/C3A), affirming the cytotoxic responses of TDCPP at higher concentrations (10–400 μg/mL), especially after long duration (2–6 days) of exposure [40,63,64,65]. Distinct cytotoxic responses of TDCPP in different cell lines have been linked with the existence of contrasting pathways in cells to safeguard them from exogenous stress [40]. Apart from the cell viability responses, morphological alterations in the cell structure are also a key aspect to determine the cytotoxicity [66]. Compared to the cobblestone and spindle structure of cells in the control, TDCPP-treated cells exhibited irregular morphology and appeared as floating bodies at higher concentrations. Our data are consistent with similar effects in TDCPP (200–400 μg/mL)-treated human keratocyte (HaCaT) and corneal epithelial (HCECs) cells exhibiting loss of their polygonal morphology and turning into round bodies, indicating the onset of apoptosis into them [40,66]. Consequently, we considered quantifying apoptosis in HepG2 cells. We have found a conspicuous increment in subG1peak in TDCPP-exposed cells, which confirms that the cytotoxicity observed in HepG2 cells is the outcome of apoptosis in HepG2 cells. Our cell cycle data are in line with previous findings affirming apoptosis induction by TDCPP in SH-SY5 cells [41], and G1 arrest in HaCaT and RAW264.7 macrophages [40,67].

Viewing the apoptotic responses in TDCPP-exposed HepG2 cells, we further advanced our analysis on the quantification of DNA damage owing to the fact that such responses are the precursors of apoptotic cell death in multicellular organisms [68]. HepG2 cells demonstrated a significant increase in the comet tail lengths, which unequivocally confirmed the genotoxic potential of TDCPP. A similar response has been reported in RAW264.7 macrophages cultured in the presence of TDCPP [67]. Intracellular ROS generation not only triggers oxidative stress in cells; rather, it also acts as a proapoptotic factor [69]. ROS production takes place in the mitochondrial respiratory chain and simultaneously affects mitochondrial homeostasis, which orchestrates apoptosis and proapoptotic signals [70]. In this context, our flow cytometric data exemplified the intracellular ROS and NO buildup, affirming oxidative stress, which simultaneously declined the MMP in TDCPP-exposed HepG2 cells. We have also found a greater level of intracellular Ca^++^ influx in the exposed cells. An elevated level of intracellular Ca^++^ influx has the ability to either directly activate caspases or cause Ca^++^ flux in mitochondria [71]. Our data on ROS, MMP, and Ca^++^ flux are in close agreement with a previous report affirming oxidative stress, mitochondrial dysfunction, and Ca^++^ flux in SH-SY5Y cells after TDCPP exposure [41]. HepG2 cells treated with TDCPP revealed a massive increase in the intracellular esterase activity. Greater fluorescence of CFDA-SE in flow cytometric analysis ratifies that fewer cell divisions take place in TDCPP-exposed cells [72].

To understand the mechanism of apoptosis, translational expressions of P53, caspase 3 and 9 proteins were analyzed in the TDCPP-exposed cells. Immunostaining analysis validated the cytoplasmic localization of the above proteins, affirming mitochondrial-dependent apoptosis in HepG2 cells. P53 is a DNA damage response protein, which gets stimulated and translocated to the nucleus; thereby, it activates sequence-specific transcription of proapoptotic and cell cycle-regulating genes [73,74]. On the other hand, caspase 9, which is an initiator caspase, gets stimulated upon disturbances in the mitochondrial integrity and cleaves the effector caspase 3 to trigger apoptosis [75]. RAW264.7 macrophages and HCECs cells exposed to TDCPP also showed the activation of caspase 3 and 9 activity to induce apoptosis [67].

We have quantitated cancer pathway genes by a qPCR array to draw a molecular understanding of TDCPP as a precursor of carcinogenesis. *GADD45G* was maximally upregulated among the cohort of 11 upregulated genes. *GADD45A* is a key regulator to sense genotoxic and non-genotoxic stress in cells. Notwithstanding the fact, it is also involved in *ΔΨm*, DNA damage, cell cycle arrest, DNA repair, and apoptosis [76]. Upregulation of *GADD45A* substantiates our comet assay, flow cytometric *ΔΨm* and apoptosis data in HepG2 cells. In line with our findings, *GADD45A* expression was upregulated in HUVECs upon exposure to a flame retardant [77]. *IGFBP5* was also upregulated in HepG2 cells after TDCPP exposure, which is in line with the overexpression of *IGFBP5* during the differentiation and apoptosis of mammary epithelial cells [78]. In addition, *GADD45* has also been involved in the overexpression of *IGFBP5* in rat kidney cells [79]. *IGFBP5* overexpression verified the possibility that proliferation of cells has been affected, and cellular senescence has been induced to initiate the apoptotic process in cells [80]. It has been validated earlier that *IGFBP5* is the inducer of cellular senescence via the p53-dependent pathway [81]. This connection is unequivocally validated by our immunofluorescence data, which have shown the localization of P53 protein in TDCPP-treated HepG2 cells, affirming cell death by cellular senescence; however, this aspect needs further evaluation in detail in future studies. TDCPP-treated cells demonstrated overexpression of *PFKL*, a prominent regulator of glycolysis. *PFKL* acts as a switch for glycolysis in rapidly multiplying cancer cells, and its upregulation boosts the proliferation and metastatic events through the activation of the Warburg effect [82]. *PFKL* upregulation in TDCPP-treated HepG2 cells implies metabolic irregularities possibly due to a larger requirement for glucose in cells, as also evident from the cytotoxicity data.

TDCPP-exposed cells exhibited the downregulation of the *SNAI1* gene, which not only codes for the zinc finger proteins but also, relatively, acts as a transcriptional repressor to safeguard the cells from apoptotic cell death [83,84,85]. Nonetheless, massive apoptosis in TDCPP-treated HepG2 cells somehow affirms the collapse of *SNAI1* function to rescue the cells. On the other hand, *TEP1* expression was also downregulated in TDCPP-treated HepG2 cells. *TEP1* is a regulator of the PI3K/Akt cell survival pathway through the dephosphorylation of PIP3 protein. Inhibition of *TEP1* leads to the activation of transcription factor (FOXO3a) to activate apoptotic cell death, as is also evident from our flow data [86,87,88]. *TEP1* downregulation has been related with miR-380-5p-mediated accumulation of P53 in diffuse malignant peritoneal mesothelioma (DMPM) cells [89]. In this line, cytoplasmic and nucleolar localization of P53 in TDCPP-treated HepG2 cells implies the likelihood of the comparable mechanism, which deserves an in-depth investigation.

## 5. Conclusions

HepG2 cells grown in the presence of TDCPP demonstrated cytotoxicity in MTT and NRU assays. Both assays affirm the fact that TDCPP acted as a toxicant, which disturbed the mitochondrial metabolism, particularly the function of mitochondrial dehydrogenase, and concurrently damaged the lysosome membrane. In comet assay analysis, TDCPP demonstrated its genotoxic capability via interacting with HepG2 nuclear DNA and introducing strand breaks into them. Besides, TDCPP-treated cells showed elevated levels of ROS and NO, which may have affected the mitochondrial integrity via alteration of membrane potential (*ΔΨm*), and simultaneously increasing the intracellular Ca^++^ level. TDCPP also acted as an antiproliferative agent, affirmed by the greater level of esterase in the exposed cells. It is obvious that TDCPP severely affected the cellular functions and its architecture, which were beyond the mitigating capacity of cellular machineries; ultimately, mitochondrial-dependent apoptotic cell death has taken place. Moreover, TDCPP demonstrated its carcinogenic attributes via the activation of prominent genes belonging to the human cancer pathway; however, validation of TDCPP carcinogenicity in suitable in vitro and in vivo test systems is warranted. Our study unequivocally substantiates the fact that TDCPP is a genotoxic, hepatotoxic, and putative carcinogenic OPFR. 

## Figures and Tables

**Figure 1 cells-11-00195-f001:**
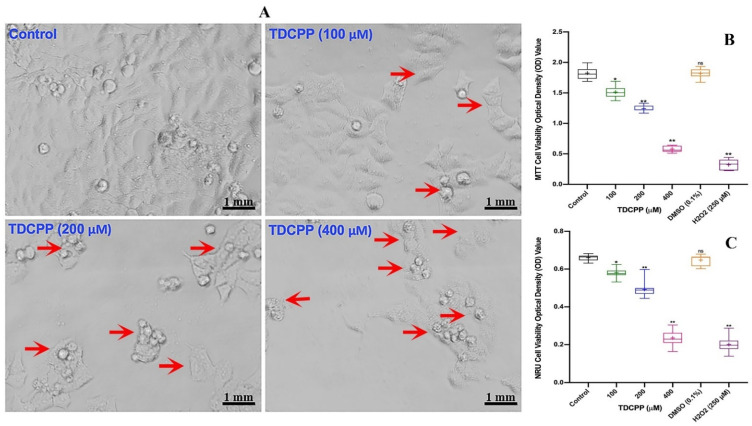
TDCPP-induced cytotoxicity in HepG2 cells. (**A**) Morphological alterations in HepG2 cells after 3 days of TDCPP exposure. Images captured at 20× magnification. (**B**) Cell survival analysis by MTT and (**C**) NRU assays. Box and whisker graphs are plotted from the optical density (OD) of three experiments performed in triplicate wells (+indicates mean value). Arrows indicate the morphological changes and dead cells. DMSO (0.1%) and H_2_O_2_ (250 μM) are negative and positive controls. * *p* < 0.05, ** *p* < 0.01 versus control. ns = non-significant versus control.

**Figure 2 cells-11-00195-f002:**
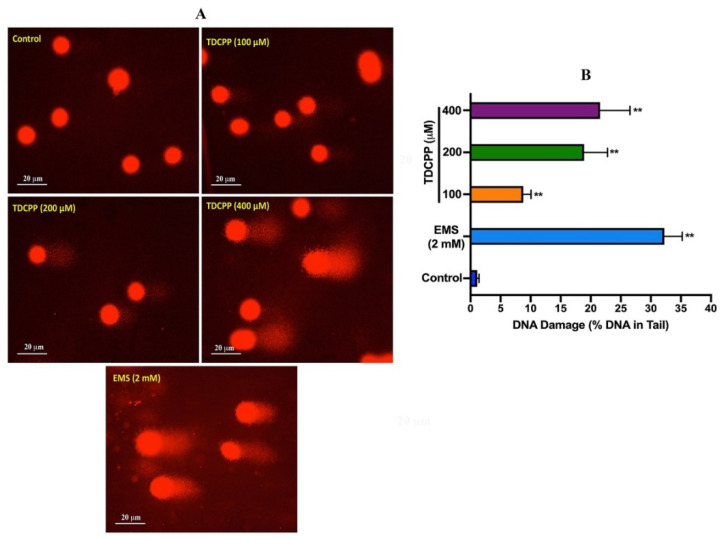
DNA damage analysis in HepG2 cells by comet assay. (**A**) Single-cell gel electrophoresis images depicting the release of broken DNA from HepG2 cells after 3 days of treatment with TDCPP. Positive control, ethyl methane sulfonate (EMS). Images were captured at 20× magnification. (**B**) DNA damage measured in terms of %DNA in the tail using comet software. ** *p* < 0.01.

**Figure 3 cells-11-00195-f003:**
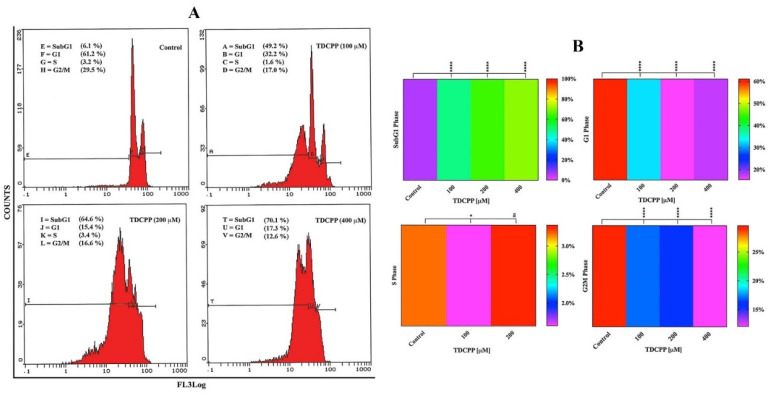
Cell cycle dysfunction in HepG2 cells after TDCPP exposure for 3 days. (**A**) Typical cell cycle images exhibiting an increase in the percentage of apoptotic cell populations in SubG1 phase post TDCPP exposure. (**B**) Mean data of cell cycle phases determined from three independent experiments performed in triplicate wells are plotted as heat maps and their significance was determined with the control (***** p <* 0.0001, * *p* < 0.012 versus control).

**Figure 4 cells-11-00195-f004:**
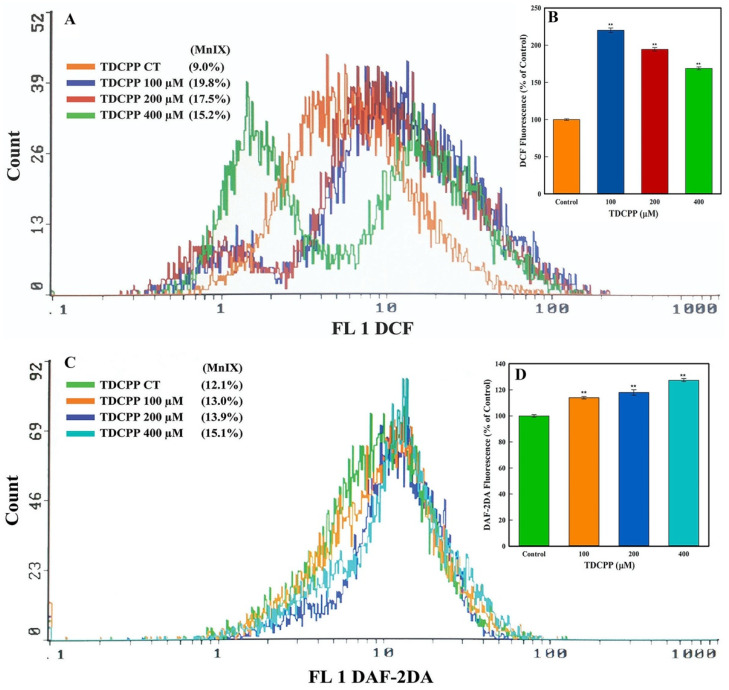
Oxidative stress quantification in TDCPP-exposed cells. (**A**) ROS and (**C**) NO measurements by flow cytometer display shift in the fluorescence spectra of DCF and DAF-2DA in TDCPP-treated cells, indicating higher ROS and NO production in HepG2 cells. MnIX: mean intensity of dye on the *x*-axis. Data sets in subfigures (**B**,**D**) are the mean ± SD of MnIX obtained from three independent experiments performed in triplicate wells (** *p* < 0.01 versus control).

**Figure 5 cells-11-00195-f005:**
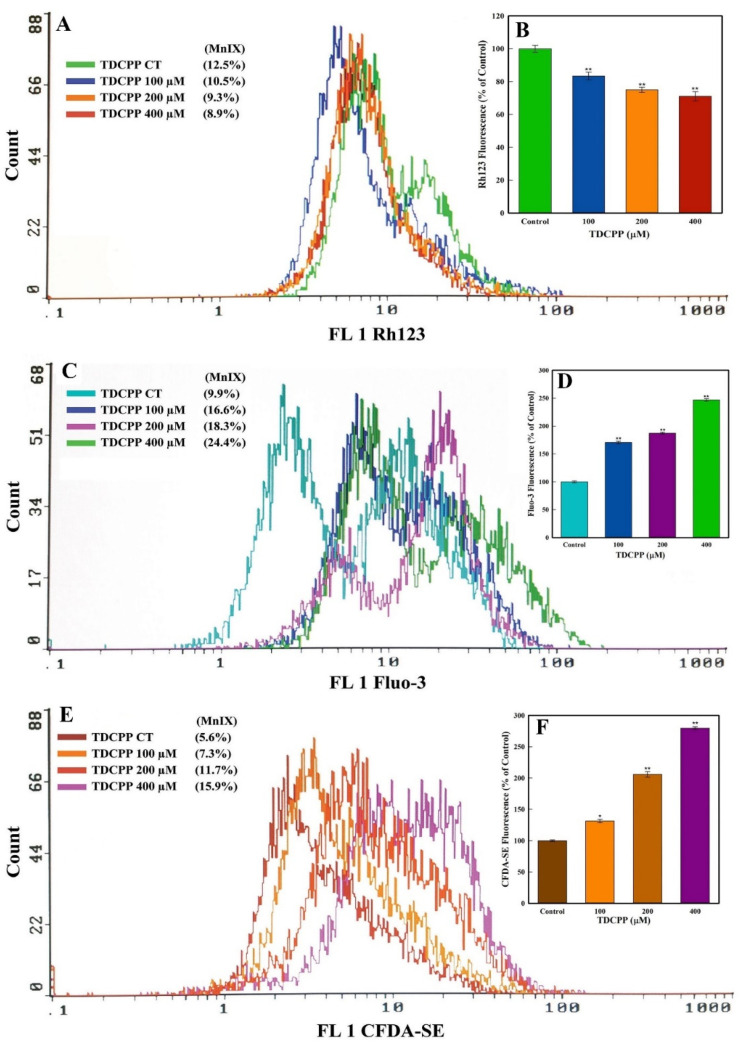
MMP (*ΔΨ*), Ca^++^ influx, and esterase quantification in HepG2 cells grown in the presence of TDCPP for 3 days. (**A**) MnIX of Rh123 dye in flow cytometric analysis depicts a fluorescence decline after TDCPP treatment, indicating the dysfunction of MMP (*ΔΨ*) in cells. (**C**,**E**) Fluo-3 and CFDA-SE dye MnIX showed greater fluorescence in TDCPP-exposed cells, indicating Ca^++^ influx and higher level of esterase. Data sets in panels (**B**,**D**,**F**) are the mean ± SD of MnIX obtained from three independent experiments performed in triplicate wells (** p <* 0.05, ** *p* < 0.01 versus control).

**Figure 6 cells-11-00195-f006:**
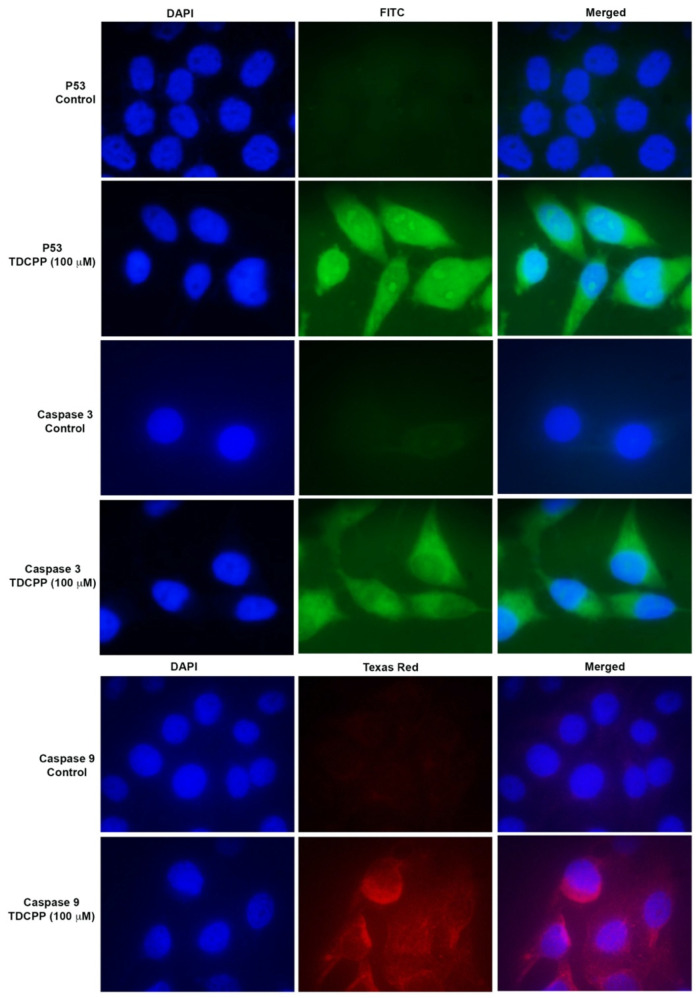
Immunofluorescence staining of P53 and caspases 3 and 9 after TDCPP (100 μM) exposure, exhibiting translational activation of DNA damage and apoptotic proteins in HepG2 cells. Images were captured at 100× magnification.

**Figure 7 cells-11-00195-f007:**
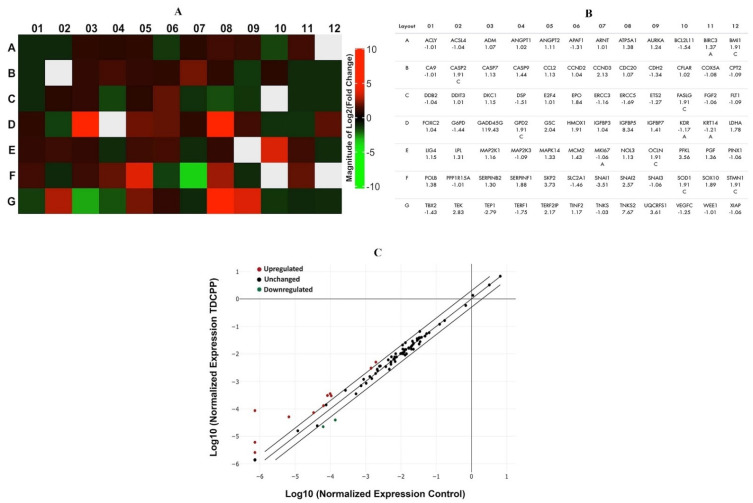
qPCR array of human cancer pathway genes in TDCPP-exposed cells. (**A**) Heat map depicting upregulated (red boxes), downregulated (green boxes), and gray boxes (undetermined) genes in the array. (**B**) Name of genes are explained according to their well location and fold regulation. (**C**) Scatter plot displays the total number of genes that exceeded the limit of >2.0-fold regulation.

## Data Availability

Not applicable.

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
