# Peer review of "Organophosphorus Flame Retardant TDCPP Displays Genotoxic and Carcinogenic Risks in Human Liver Cells"

_cells, 2022, doi:10.3390/cells11020195_

Round 1

Reviewer 1 Report

Saquib et. al. has taken human liver cells (HepG2) for studying the hepatotoxicity of OPFRs. Specifically, they have selected TDCPP, owing to the reality that it has not been studied for genotoxicity and its role as a putative carcinogen in human liver cells. Authors have measured the cytotoxicity and quantitated DNA damage by comet assay. Using the flow cytometry authors have measured the oxidative stress and mitochondrial dysfunction in HepG2 after 3 days of TDCPP treatment. HepG2 cells grown in the presence of TDCPP demonstrated cell death, analyzed by flow cytometry. Transcriptomic alterations in human cancer pathway were observed in HepG2 after TDCPP exposure using qPCR array experiments. 

This study is carefully planned, and executed in a logical way to provide ample evidences on the genotoxic nature of TDCPP, as well as indicating that it may act as a carcinogenic agent, which may trigger health consequences in humans. Current work is timely in reporting the toxicity of TDCPP in liver cells, which may act a potent toxicant to humans. However, authors should incorporate major changes in their manuscript. 

Major comments;

--Authors have selected high concentrations to expose the cells, please justify and insert this justification in the manuscript.

--Lines 97-100. Authors stated that they have screened TDCPP cytotoxicity between 4-500 µM for 24 and 48h, add this data in supplementary file. 

--Figure 1, mark the apoptotic/dead cells with arrows.

-- Since the adverse effects are associated with ROS, has the authors performed any experiments to see if the effects can be abrogated by antioxidants?

-- Although the manuscript is generally readable, authors are suggested to thoroughly check the manuscript for syntax errors.

Overall, I recommend the acceptance of manuscript after major revision.

Author Response

Reviewer #1

Saquib et. al. has taken human liver cells (HepG2) for studying the hepatotoxicity of OPFRs. Specifically, they have selected TDCPP, owing to the reality that it has not been studied for genotoxicity and its role as a putative carcinogen in human liver cells. Authors have measured the cytotoxicity and quantitated DNA damage by comet assay. Using the flow cytometry authors have measured the oxidative stress and mitochondrial dysfunction in HepG2 after 3 days of TDCPP treatment. HepG2 cells grown in the presence of TDCPP demonstrated cell death, analyzed by flow cytometry. Transcriptomic alterations in human cancer pathway were observed in HepG2 after TDCPP exposure using qPCR array experiments.

This study is carefully planned, and executed in a logical way to provide ample evidences on the genotoxic nature of TDCPP, as well as indicating that it may act as a carcinogenic agent, which may trigger health consequences in humans. Current work is timely in reporting the toxicity of TDCPP in liver cells, which may act a potent toxicant to humans. However, authors should incorporate major changes in their manuscript.

Author Response

We sincerely thank reviewer for the critical analysis of our manuscript and appreciating words. As suggested, the manuscript has been revised considering your suggestions and queries. Corrections are highlighted and done in track change mode.

Major Comment 1

--Authors have selected high concentrations to expose the cells, please justify and insert this justification in the manuscript.

Author Response Major Comment 1

Reviewer remark on concentration selection is greatly appreciated. For your kind information, exposure concentrations of TDCPP were selected based on the screening tests done for 24-72h with a range of concentrations (5, 10, 25, 50, 100, 200, 400 µM). However, upto 50 µM no significant cytotoxic effects were observed till 72h. Only higher concentrations (i.e 100, 200, 400 µM) demonstrated significant cytotoxicity after 72 h. Therefore, these concentrations were selected for further experiments.

Above justification on the selection of TDCPP exposure concentrations is now inserted in the revised manuscript.

Please see highlighted texts on page 3 lines 112-116 in the revised manuscript.

Major Comment 2

--Lines 97-100. Authors stated that they have screened TDCPP cytotoxicity between 4-500 µM for 24 and 48h, add this data in supplementary file.

Author Response Major Comment 2

As desired, a supplementary figure is now added in the manuscript for the screening experiments data.            

Please see revised supplementary figure 1.

Major Comment 3

--Figure 1, mark the apoptotic/dead cells with arrows.

Author Response Major Comment 3

As suggested, marking of apoptotic/dead cells are now done.        

Please see revised Figure 1 in the manuscript on page 5.

Major Comment 4

-- Since the adverse effects are associated with ROS, has the authors performed any experiments to see if the effects can be abrogated by antioxidants?

Author Response Major Comment 4

We appreciate reviewers query on the abrogation of ROS by antioxidants. In this study, we have focused to evaluate the role of TDCPP to act as an oxidative stressor by measuring ROS. Hence, we did not quantitate the effects of antioxidants on the suppression of ROS/oxidative stress in cells. Indeed, it is a valuable suggestion; however, we are unable to measure such change due to restricted lab access in the pandemic. We hope that our learned reviewer will understand our limitation.   

No changes.

Major Comment 5

-- Although the manuscript is generally readable, authors are suggested to thoroughly check the manuscript for syntax errors.

Author Response Major Comment 5

Reviewer suggestion on the English check is appreciated. The manuscript is now thoroughly checked for grammatical mistakes.         

Please see revised manuscript.

Reviewer 2 Report

Manuscript Title: Organophosphorus Flame Retardant TDCPP Displayed Its 2 Genotoxic and Carcinogenic Risks in Human Liver Cells.

In my opinion this is an interesting manuscript, addressing an important question. I reckon that this article would be interesting for Readers of Cells.

Some minor items to improve the manuscript:

  1. The concentrations ranges of the compound studied detected in marine and freshwater animals, birds, insects, and human samples should be added. This information would improve the manuscript and would be useful for the Readers.
  2. A concentrations range of TDCPP used in the experiments was 100-400 μM. What made the Authors choose such concentrations? How can the tested concentrations range be related to the concentrations determined in biological samples/environment?
  3. Please, describe the principles of determination in the methods used.
  4. There is an inexpressible scale in Figure 2.
  5. I propose to add to figure 2 a graph showing the statistical changes under the influence of TDCPP. X (DNA damage [% DNA in tail] ) and Y – concentration of TDCPP.
  6. In addition to this, the respected Authors are requested to cite the following reference: Bukowski, K.; Wysokinski, D.; Mokra, K.; Wozniak, K. DNA damage and methylation induced by organophosphate flame retardants: Tris(2-chloroethyl) phosphate and tris(1-chloro-2-propyl) phosphate in human peripheral blood mononuclear cells. Human Exp. Toxicol., 2019, 38(6), 724–733. doi: 10.1177/0960327119839174.

Author Response

Reviewer #2

In my opinion this is an interesting manuscript, addressing an important question. I reckon that this article would be interesting for Readers of Cells.

Comment 1

The concentrations ranges of the compound studied detected in marine and freshwater animals, birds, insects, and human samples should be added. This information would improve the manuscript and would be useful for the Readers.

Author Response to Comment 1

Our sincere thanks to the reviewer. As suggested, we have added the indicated information’s on the occurrence of TDCPP in different organisms.             

Please see page 1 line 45, page 2 lines 46-50, 55-63 in the revised manuscript.

 Comment 2

A concentrations range of TDCPP used in the experiments was 100-400 μM. What made the Authors choose such concentrations? How can the tested concentrations range be related to the concentrations determined in biological samples/environment?

Author Response to Comment 2

Reviewer’s remark on concentration selection is greatly appreciated. For your kind information, the exposure concentrations of TDCPP were selected based on the screening tests done for 24-72h with range of concentrations (5, 10, 25, 50, 100, 200, 400 µM). However, upto 50 µM no significant cytotoxic effects were observed till 72h. Only higher concentrations (i.e 100, 200, 400 µM) demonstrated significant cytotoxicity after 72 h. Therefore, these concentrations were selected for further experiments. Above justification on the selection of TDCPP exposure concentrations is now inserted in the revised manuscript.

Concerning the environmental relation of tested concentrations in our study, it reflects the existence of realistic conditions of TDCPP concentrations in indoor dust ranges from <0.03 to 326 mg/g (Carignan et al. 2013). Previous studies on TDCPP also verified its toxicity in mammalian cells ranging from 0 to 129 µg/ml (Cui et al. 2020), and ≥100 μg/mL for 48 h (Killilea et al. 2017); relatively, we have used 100 µM (=58 µg/ml), 200 µM (=116 µg/ml) and 400 µM (=232 µg/ml) in current study.   For a better clarity to readers, we have inserted the statement on the selection of exposure concentrations in the revised manuscript.

Please see highlighted texts on page 3 lines 112-119 in the revised manuscript.

Comment 3

Please, describe the principles of determination in the methods used.

Author Response to Comment 3

As suggested, principles of determination in the methods are added in the revised manuscript.

Please see page 3 lines 119-121, 128-130, 138-140; page 4 lines 152-154; lines 173-175, 184-185 in the revised manuscript.

Comment 4

There is an inexpressible scale in Figure 2.

Author Response to Comment 4

Figure 2 is revised and the scale is now clear.          

Please see revised figure 2.

Comment 5

I propose to add to figure 2 a graph showing the statistical changes under the influence of TDCPP. X (DNA damage [% DNA in tail] ) and Y – concentration of TDCPP.

Author Response to Comment 5

As desired, we have modified figure 2 and a new panel is added describing X (DNA damage [% DNA in tail]) and Y – concentration of TDCPP.

Please see panel II in the revised figure 2.

Comment 6

In addition to this, the respected Authors are requested to cite the following reference: Bukowski, K.; Wysokinski, D.; Mokra, K.; Wozniak, K. DNA damage and methylation induced by organophosphate flame retardants: Tris(2-chloroethyl) phosphate and tris(1-chloro-2-propyl) phosphate in human peripheral blood mononuclear cells. Human Exp. Toxicol., 2019, 38(6), 724–733. doi: 10.1177/0960327119839174.

Author Response to Comment 6

The indicated reference is added in the comet assay method.        

Please see reference # 58.

Reviewer 3 Report

The study is a valuable source of information about genotoxic and carcinogenic effect of  organophosphorus compounds in everyday use products. However, the manuscript requires some minor revision:

  1. Abbreviations in Materials and Methods should be explained, eg. DMEM
  2. Positive and negative MTT must be added and described.
  3. Standard deviation must be added to viability values.
  4. EC values should be added.
  5. In MTT assay a recommended use box and whiskers graph plus asterisk, instead the columnar ones.

After minor revision, I recommend that you accept manuscript for publication.

Author Response

Reviewer #3

The study is a valuable source of information about genotoxic and carcinogenic effect of  organophosphorus compounds in everyday use products. However, the manuscript requires some minor revision:

Comment 1

Abbreviations in Materials and Methods should be explained, eg. DMEM

Author Response to Comment 1

Thank you very much for your valuable comments. As indicated, abbreviations in the manuscript are explained at their first appearance.      

Please see page 1 lines 13, 17-18; page 3 lines 124-125 in the revised manuscript.

Comment 2

Positive and negative MTT must be added and described.

Author Response to Comment 2

 As indicated positive and negative controls are added in the MTT assay and described.   

Please see revised figure 1 on page 5, lines 212-216.

Comment 3

Standard deviation must be added to viability values.

Author Response to Comment 3

As suggested, SD values are now added to viability values.

Please see page 4 lines 201-202; page 5 lines 204-205 in the revised manuscript.

Comment 4

EC values should be added.

Author Response to Comment 4

Reviewer remark on EC is greatly appreciated. We have determined the IC50 of TDCPP on HepG2, which is 163.2 μM, LogIC50 2.213, R2 0.9165. The above information is now added in the revised manuscript.     

Please see page 4 lines 202-203 in the revised manuscript. 

Please see revised figure 2.

Comment 5

In MTT assay a recommended use box and whiskers graph plus asterisk, instead the columnar ones.

Author Response to Comment 5

As desired, we have revised figure 1 and replaced the histogram plots with recommended whiskers graph (with asterisks).     

Please see revised figure 1 panels II and III.

Round 2

Reviewer 1 Report

Article can be accepted.